# Reverse Shoulder Arthroplasty Biomechanics

**DOI:** 10.3390/jfmk7010013

**Published:** 2022-01-19

**Authors:** Christopher P. Roche

**Affiliations:** Exactech, Inc., Gainesville, FL 32653, USA; chris.roche@exac.com

**Keywords:** reverse total shoulder arthroplasty, biomechanics, shoulder

## Abstract

The reverse total shoulder arthroplasty (rTSA) prosthesis has been demonstrated to be a viable treatment option for a variety of end-stage degenerative conditions of the shoulder. The clinical success of this prosthesis is at least partially due to its unique biomechanical advantages. As taught by Paul Grammont, the medialized center of rotation fixed-fulcrum prosthesis increases the deltoid abductor moment arm lengths and improves deltoid efficiency relative to the native shoulder. All modern reverse shoulder prostheses utilize this medialized center of rotation (CoR) design concept; however, some differences in outcomes and complications have been observed between rTSA prostheses. Such differences in outcomes can at least partially be explained by the impact of glenoid and humeral prosthesis design parameters, surgical technique, implant positioning, patient-specific bone morphology, and usage in humeral and glenoid bone loss situations on reverse shoulder biomechanics. Ultimately, a better understanding of the reverse shoulder biomechanical principles will guide future innovations and further improve clinical outcomes.

## 1. Reverse Shoulder Arthroplasty: Form and Function

The reverse total shoulder arthroplasty (rTSA) prosthesis is geometrically different than the native glenohumeral joint, and these differences in form substantially modify how the shoulder mechanically functions. The rTSA “reverses” the native articular concavities of the glenoid and humerus such that a concave humeral prosthesis is positioned below a convex, spherical glenoid prosthesis. Positioning the humerus below a spherical glenoid provides the rTSA with several biomechanical advantages relative to the native shoulder, but also a few disadvantages.

The first biomechanical advantage of rTSA is that the spherical glenoid component, called a glenosphere, acts as a mechanical stop for the humerus, preventing humeral head superior migration during deltoid contraction. The deltoid is the largest muscle in the shoulder and consists of three heads that insert on the lateral side of humeral diaphsysis: (1) anterior deltoid (originating from the anterior acromion and clavicle); (2) middle deltoid (originating from the lateral margin of the acromion); (3) the posterior deltoid (originating from the scapular spine). Deltoid contraction primarily powers arm elevation. This mechanical stop converts the deltoid force into rotation, and in doing so, functionally restores joint stability for patients with an unstable shoulder. 

The second biomechanical advantage of rTSA is that the glenosphere functions as a fixed mechanical fulcrum, converting the superiorly directed pull of the deltoid into arm elevation and rotation, thereby enabling active arm elevation in multiple planes over a wide range of motion. Reversing the articular concavities with rTSA is a functional substitute for the concavity compression stability mechanism that occurs in the native shoulder by the rotator cuff pulling the convex humeral head articular surface into the concave glenoid articular surface. This coordinated action of the rotator cuff muscles converts the superiorly directed pull of the deltoid into arm elevation and rotation [1]. The rotator cuff is composed of four muscles (supraspinatus, subscapularis, infraspinatus, and teres minor) that surround the proximal humeral head and function together as a dynamic fulcrum in the native shoulder to pull the humeral head into the center of the glenoid of the scapula during all positions of arm motion [1,2,3,4,5,6]. While some or all of the rotator cuff muscles may be functional with rTSA, concavity compression is no longer necessary for arm elevation with rTSA because of the fixed mechanical fulcrum and the work of the deltoid. However, a nonfunctioning deltoid is a contraindication of rTSA.

The third biomechanical advantage of rTSA is the increased deltoid abductor moment arm lengths due to the medial and inferior shift of the joint center of rotation (CoR) relative to the native shoulder (Figure 1). rTSA prostheses are typically associated with 5–10 mm [7,8,9,10] inferior and 20–30 mm [8,9,11,12,13,14] medial translation of the CoR relative to the native shoulder. Medially translating the CoR is biomechanically beneficial because it increases the length of the deltoid abductor moment arm from 10–30 mm [15,16,17,18,19] for the native shoulder with the arm at the side to 22–40 mm [19,20,21,22] for the rTSA construct (Figure 2). Muscles generate straight-line forces that are converted to torques in proportion to their perpendicular distance between the joint CoR and the muscle’s line of action [15,23]. This perpendicular distance is called the muscle moment arm, and the greater the muscle’s moment arm, the greater its capacity to generate the torque required for motion and to support external loads. For this reason, increasing the abductor moment arm length with rTSA improves the efficiency of the deltoid by requiring less muscle force to generate the same amount of torque.

Unfortunately, reversing the concavities with rTSA introduces a few biomechanical disadvantages relative to the native shoulder. As described in Figure 1, positioning the rTSA humeral component below the glenosphere both medially and inferiorly translates the humerus relative to the native shoulder. rTSA prostheses are typically associated with a 25–40 mm inferior [8,9] and 5–20 mm medial [8,9,24] translation of the humerus relative to the native shoulder. Medially translating the humerus is biomechanically unfavorable because it shortens the rotator cuff muscles and limits their ability to generate active internal/external rotation [8,9,24,25]. Inferiorly translating the humerus is biomechanically unfavorable because it alters the native orientation of the humeral muscle insertions relative to the CoR. Doing so changes each muscle’s moment arms and muscle lengths and modifies how each muscle influences motion relative to its native physiologic function. Specifically, inferior humeral translation elongates the deltoid beyond its native length, and the associated tension has been cited [26,27] as the reason for acromial and scapular stress fractures after rTSA, a serious rTSA-specific complication for which there is no currently accepted optimal treatment solution. Combined, inferiorly and medially translating the humerus has been demonstrated to elongate the deltoid by as much as 20% with some rTSA prosthesis designs, [9] reduce deltoid wrapping around the greater tuberosity, which has negative implications on joint stability and deltoid moment arm length [8,9], reduce range of motion, and shorten the rotator cuff muscles by as much as 40% [8,9]. Shortening of the rotator cuff may also be responsible for the modified scapulohumeral rhythm reported by Walker et al. [28] with rTSA, in which a greater proportion of scapular rotation occurs relative to the native shoulder (Figure 3).

## 2. Grammont Reverse Shoulder

Reeves et al. [29] was the first to publish on a reverse shoulder prosthesis in 1972; however, there were several other reverse shoulder designs developed in the early 1970s and it is unclear who first invented the concept. Charles Neer reported developing the Mark 1, Mark 2, and Mark 3 reverse shoulder prostheses between 1970 and 1973, [30,31] but other early reverse shoulder designs were also implanted in the 1970s, including the Kolbel, Fenlin, Reeves-Leeds, Gerard and Lannelong, and Kessel reverse shoulders [29,30,31,32,33,34,35,36]. All of these reverse shoulder prostheses were constrained and utilized a fixed-fulcrum concept, securing a spherical glenoid to the scapula with a post so that it was laterally offset from the glenoid fossa. None of these early reverse shoulder designs were clinically successful, and as a result, all were abandoned in the US and European markets [31]. 

In 1985, Paul Grammont developed a new concept for a reverse shoulder prosthesis. His initial design featured a glenoid component that was two-thirds of a sphere fixed directly on the glenoid; this is in contrast to historical designs which utilized a sphere on a post. This new concept for a reverse shoulder prosthesis was revolutionary because it medialized the CoR relative to the native shoulder [12,37,38]. In 1991, Grammont developed the Delta III and refined his medialized CoR fixed-fulcrum concept by utilizing a glenosphere geometry having an equivalent articular radius and thickness (i.e., a hemisphere), which further medialized the CoR and positioned it at 0 mm (e.g., directly on the glenoid fossa). By medializing the CoR to the glenoid fossa, Grammont further increased the deltoid abductor moment arm lengths and improved deltoid efficiency [12,39]. Medializing the CoR had an additional benefit in that it also reduced the torque on the glenoid bone–implant interface, thereby reducing the risk of aseptic glenoid loosening. Other important design features utilized by the Delta III include a humeral prosthesis that was inlayed in the proximal humerus, a 155° humeral neck–shaft inclination angle, and a circular glenoid baseplate having a central peg with divergent fixation screws. 

Since its release in 1991, the Delta III has been used to successfully treat numerous end-stage degenerative conditions of the shoulder for which there was no previous solution [12,39]. Because of these clinical successes, the Delta III was 510k cleared in the US market in November 2003. Since that 510k clearance, numerous rTSA prostheses have been developed to further improve rTSA performance and also mitigate the occurrence of complications. The literature has demonstrated that different rTSA prostheses are associated with (1) differences in the amount of humeral and glenoid bone removed during implantation [40], (2) differences in overall glenoid fixation and the methods to achieve fixation [41,42,43,44,45,46,47,48], (3) differences in ranges of motion, including different locations of impingement and different rates of scapular notching [49,50,51,52,53,54,55], and (4) differences in CoR and humeral positioning relative to the native shoulder, which directly influences several biomechanical parameters, including muscle moment arms [7,19,20,21,22], muscle lengths [8,9], and amount of deltoid wrapping around the greater tuberosity at various arm positions [8,9]. While these new design improvements and surgical technique enhancements have successfully reduced the occurrence of rTSA-related complications, improved clinical outcomes, increased range of motion, improved predictability of outcomes, and extended the use of reverse shoulder arthroplasty to new indications, including substantial glenoid and humeral bone loss, it must be acknowledge that every modern rTSA prosthesis shares Grammont’s medialized CoR heritage; this is his great innovation and it is what is most responsible for the clinical improvement observed with rTSA.

## 3. rTSA Prosthesis Design Considerations on Biomechanics

When performing a design optimization, it is important for the medical device developer to understand the unique interaction of the multiple various design parameters for a particular design problem. All design changes are associated with trade-offs, and with any design problem, there is rarely one optimal configuration for all use cases. Typically, there are numerous optimal configurations or configurations that confer benefit for some patients in some cases and for other patients in other cases. Accordingly, it is important to inform the user on how to select the optimal design for the correct use-case. For the reverse shoulder prosthesis, there are numerous design parameters to consider when performing a design optimization that selects the parameters that maximize performance while simultaneously de-selecting the parameters that predispose for higher complication risks. 

An illustrating example of a reverse shoulder optimization analysis was conducted in 2006 by Roche et al. [53,54], who used computer modeling to independent quantify the contribution of humeral neck angle, glenosphere thickness, glenosphere diameter, glenosphere inferior offset, and humeral liner constraint on impingement, range of motion, and joint stability (as measured by the jump distance, the lateral joint displacement needed for the humeral component to escape the glenosphere). From this computer model, they demonstrated that the Grammont Delta III prosthesis impinges on the inferior scapular neck at 31° humeral abduction and concluded that the specific design parameters combinations utilized in the Delta III are not optimized. As it relates to glenosphere thickness, every 1 mm that the CoR is lateralized achieves ~5° more humeral adduction prior to impingement but has no impact on jump distance. Modifying glenosphere diameter without changing the position of the CoR (by correspondingly modifying thickness) did not change overall range of motion; however, increasing glenosphere diameter by 2 mm did increase jump distance by 0.5 mm. As it relates to glenosphere inferior offset, every 1 mm inferior shift in position achieves ~4° more humeral adduction prior to impingement. On the humeral side, as it relates to humeral neck angle, every 5° decrease in humeral neck angle relative to the 155° Delta III shifts by 5° the inferior and superior scapular impingement points. To clarify, decreasing the humeral neck angle did not increase jump distance or the overall range of motion; however, it did reduce the inferior impingement during adduction at the expense of less humeral abduction. Finally, decreasing humeral liner constraint by 0.0125 increases overall range of motion by 4° but also decreases jump distance by 0.5 mm. Based upon these findings, Roche et al. [53,54] recommended several design changes for the Grammont Delta III, and in doing so, encouraged numerous subtle design modifications of each parameter to minimize the tradeoff associated with each.

With Grammont’s glenosphere, the medial/lateral position of the CoR relative to the glenoid is determined by the difference between the glenosphere thickness and the glenosphere articular radius. Nearly all reverse shoulder designs developed since the Delta III utilize a glenosphere having a diameter between 32 and 46 mm and a thickness that is slightly more than its articular radius so that they position the CoR slightly lateral to the glenoid fossa. One unique exception is the Equinoxe inset CoR glenosphere, which is a 46 mm glenosphere that has a thickness of 21 mm, so that it positions the CoR 2 mm into to the glenoid fossa (Figure 4). By further medializing the CoR into the glenoid relative to the Delta III, this unique inset CoR glenosphere can increase deltoid efficiency by up to 10.5% relative to the Delta III and increase deltoid efficiency by >40% relative to the most lateralized CoR design available on the market [19]. However, the primary reason nearly every rTSA device developed since the Delta III utilizes a glenosphere thicker than a hemisphere is because a thicker glenosphere greatly limits humeral liner impingement with the scapula neck and reduces the occurrence of scapular notching [49,50,51,52,53,54,55]. Scapular notching is a unique complication introduced by Grammont’s medialized CoR concept. It has been demonstrated to be progressive and extend beyond the inferior baseplate screw [56,57,58], as described in the Grade 4 Nerot-Sirveaux58 classification (suggesting that an osteolytic response can also occur in response to the polyethylene wear debris). Scapular notching has been reported to occur in as many as 96% of Grammont Delta III reverse shoulders [58]. Scapular notching has been demonstrated to compromise glenoid baseplate fixation [45] and is also associated with poorer clinical outcomes, less range of motion, and less strength [59,60,61]. 

Lateralizing the CoR reduces humeral liner impingement with the scapular neck [49,50,51,52,53,54,55]. However, lateralizing the CoR also increases the torque on the glenoid bone–implant interface [45,46,48] and decreases the length of the deltoid abductor moment arms [7,19,20,52,62,63] (Figure 4). As the deltoid abductor moment arms are decreased, the deltoid becomes less effective as an abductor and requires a greater force to elevate the arm. Additionally, these elevated muscle loads increase scapular bone stresses, thereby increasing the risk of acromial and scapular insufficiency fractures, which occur at a higher rate in rTSA prostheses that lateralize the CoR relative to other rTSA design styles, as demonstrated by a recent literature meta-analysis by King et al. [64]. 

Modifying Grammont’s inlay humeral prosthesis to increase lateralization through the humerus has been proposed as an alternative method to maintain Grammont’s deltoid abductor moment arm lengths while also lateralizing the CoR to reduce the occurrence of humeral liner–scapular neck impingement [53,54,65]. Humeral lateralization can be accomplished by decreasing the humeral neck angle relative to Grammont 155°, decreasing the humeral liner constraint, and/or increasing the offset of the humeral liner–humeral stem connection. However, the most effective method to achieve joint lateralization without lateralizing the CoR is through the use of an onlay humeral prostheses. By placing the humeral tray/liner on top of the anatomic neck osteotomy, the humerus is shifted both laterally and inferiorly relative to Grammont Delta III, though it is important to note that the humerus is still more medial relative to the native joint and therefore it still shortens the rotator cuff relative to its native length. However, this relatively more lateral humeral positioning better tensions the residual rotator cuff muscle, offering potential for improved active internal/external rotation. More medialized rTSA prostheses are associated with greater rotator cuff muscle shortening, which limits their ability to generate active internal/external rotation and impairs a patient’s ability to conduct several activities of daily living. This more lateral positioning with an onlay humeral rTSA prosthesis lengthens the deltoid moment arm to improve joint efficiency and also better restores deltoid wrapping around the greater tuberosity. Greater deltoid wrapping offers the potential for improved joint stability [66,67,68,69]. More medialized reverse shoulder prostheses are associated with less deltoid wrapping, which reduces the horizontal stabilizing compressive force vector of the deltoid and may increase the risk of dislocation, particularly in cases of substantial medial glenoid wear [70] (Figure 5). When the native joint line is medialized with substantial glenoid erosion, the humerus is medialized and the deltoid no longer wraps around the greater tuberosity, eliminating the compressive deltoid force vector, and with sufficient humeral medialization, the deltoid may even impose a distraction vector resulting in instability. Finally, as an additional benefit, the onlay humeral component can also function as a platform humeral stem, sharing the same connection as the humeral head when used for anatomic total shoulder arthroplasty (aTSA), which has numerous clinical advantages and inherent efficiencies, and most importantly, as reported by Crosby et al. [71], does not require removal of a well-fixed humeral stem when revising a failed aTSA to an rTSA. 

rTSA design classification systems have been proposed to objectively categorize glenoid designs based on how they position the CoR and humeral designs based on how they position the humerus, and then combine each together to account for the combined offset and interaction. In 2013, Roche et al. [65] and Hamilton et al. [20] first introduced an rTSA design classification system, which was later refined by Routman et al. [9] This design classification system categorized glenoid prostheses with a CoR of 5 mm or less from the glenoid face as medialized glenoid (MG), and glenoid prostheses with a CoR greater than 5 mm from the glenoid face as lateralized glenoid (LG) (Figure 6). Additionally, this rTSA design classification system categorized a humeral component by humeral offset. Humeral offset is defined as the horizontal distance between the intramedullary canal/humeral stem axis to the center of the humeral liner and determines the amount of humeral lateralization. Humeral offset/humeral lateralization is influenced by humeral neck angle, humeral osteotomy, and use of an inset or onset humeral tray/stem design, where an onset humeral design includes a modular humeral tray that sits on top of the resection and may or may not be offset. Humeral prostheses with an offset of 15 mm or less are medialized humerus (MH), and humeral prostheses with an offset greater than 15 mm are lateralized humerus (LH) (Figure 7). By combining the glenoid and humeral prosthesis categorizations (MG/MH, LG/MH, MG/LH, and LG/LH) (Figure 8), an overall rTSA design classification is defined that describes the reverse shoulder prosthesis position of the CoR and also humeral offset; these two parameters both independently and combined influence biomechanics. 

The MG/MH reverse shoulder prosthesis is the Grammont Delta III prosthesis and is associated with the most medial positioning of the CoR and the humerus. Because of this amount of medialization, concomitant repair of the subscapularis is recommended with this device to maintain stability [72,73]. Furthermore, because of this medialization, the use of MG/MH prostheses with an uncorrected glenoid deformity is discouraged. In such cases, bone graft or an augmented baseplate may be required to lateralize the joint line and achieve stability. Relative to the MG/MH design, an LG/MH design can utilize its more lateral glenoid position to lateralize the joint line and better tension the residual rotator cuff and also improve deltoid wrapping. By lateralizing the humerus through the glenoid, it is inherently more stable than the MG/MH prosthesis, and because of this, an LG/MH rTSA design may not require concomitant repair of the subscapularis to achieve stability [74]. However, it should be noted that the resulting deltoid abductor moment arm of the LG/MH design is less than that of MG/LH designs due to its more lateralized CoR. In contrast, an MG/LH rTSA design can utilize its more lateral humeral position to compensate for the relative joint medialization caused by the MG while maintaining a medialized CoR, thereby better tensioning the residual rotator cuff, better restoring deltoid wrapping, and further increasing the deltoid abductor moment arms. Friedman et al. reported that concomitant repair of the subscapularis is not necessary for stability with an MG/LH rTSA design [75]. A fourth rTSA design classification is the LG/LH prosthesis. Theoretically, this design style can achieve the same or better residual rotator cuff tensioning and deltoid wrapping as an LG or LH device can; however, it will have shorter deltoid abductor moment arms relative to MG/LH designs because of its lateralized CoR. It should be noted that there are not currently any LG/LH prostheses commercially available; however, when an MG/LH design is used in combination with a BIO-RSA technique (i.e., bone grafting a noneroded glenoid to achieve lateralization) [76], it effectively functions as an LG/LH design. Care should be made to not over-tension the shoulder when performing a BIO-RSA with an MG/LH prosthesis. Similarly, using a BIO-RSA technique on an MG/MH design converts that rTSA design style to an LG/MH. 

As further evidence of biomechanical differences between these different rTSA prosthesis designs, MG/MH rTSA, LG/MH rTSA, and MG/LH rTSA, Liou et al. utilized a computer muscle model to quantify the muscle forces and joint reaction forces associated with these devices and compared those biomechanical characteristics to the native shoulder [63]. Liou et al. reported that all three rTSA prostheses demonstrated larger deltoid abductor moment arms, a decreased joint reaction force, and a decreased deltoid force during arm elevation relative to the native shoulder. Liou et al. also reported that the MG/LH rTSA prosthesis was associated with the lowest joint reaction force and lowest middle deltoid forces during both abduction and forward flexion as compared to the MG/MH and LG/MH rTSA designs. Similar findings demonstrating improved deltoid muscle efficiency with humeral lateralization were reported by both Giles et al. and Henninger et al. [7,62]. 

Within a given rTSA design classification, it is important to note that not all rTSA devices of the same design style achieve the same clinical results. For example, Routman et al. reported that 61 of 4125 patients had an acromial or scapular fracture with an MG/LH reverse shoulder prosthesis (Equinoxe; Exactech, Inc.; Gainesville, USA), for an overall fracture rate of 1.5% [77]. In contrast, Ascione et al. [78] reported that 21 of 485 patients had a scapular fracture with a different MG/LH reverse shoulder prosthesis (Ascend Flex; Stryker, Inc.; Kalamazoo, USA), for an overall fracture rate of 4.3%, and Haidamous et al. [79] reported that 10 of 84 Ascend Flex patients had a scapular fracture, for an overall fracture rate of 11.9%. While both the Ascend Flex and Equinoxe rTSA prostheses can be classified as MG/LH designs, each of these designs are associated with different amounts of humeral lateralization and distalization. As described in Figure 9 and as reported by Routman et al. [9], the Ascend Flex is associated with 5 mm greater humeral distalization and 3 mm less lateralization than that of the Equinoxe. Routman et al. [9] used a computer muscle model to compare these two MG/LH rTSA designs and demonstrated that the Ascend is associated with greater deltoid elongation, greater rotator cuff shortening, and less deltoid wrapping compared to the Equinoxe. Combined, these clinical and computer modeling results suggest that stretching the deltoid in the lateral direction is associated with a lower risk of scapular fractures than is distally stretching the deltoid when using an MG/LH design. Another important consideration when comparing these devices is that Ascione et al. [78] reported the BIO-RSA technique was used in the majority of scapular fracture cases (12 of 21) in order to achieve “glenoid lateralization”; such use of the BIO-RSA technique effectively converts the Ascend rTSA into an LG/LH design. Based upon these results, LG/LH humeral designs may be at the greatest risk for scapular fractures. 

This rTSA design classification system has recently been further refined by Werthel et al. [80] by adding one new humeral offset category: minimally lateralized humerus. Using this new offset category, Werthel et al. grouped these implants into five different combined configurations by a measurement of global offset: medialized (M) rTSA, minimally lateralized (ML) rTSA, lateralized (L) rTSA, highly lateralized (HL) rTSA, and very highly lateralized (VHL) rTSA. As a validation of this modified rTSA classification system, Werthel et al. quantified the offset of 24 different rTSA prostheses and sorted each into global offset configurations as follows: five M rTSA, five ML rTSA, seven L rTSA, six HL rTSA, and one VHL rTSA, thereby demonstrating that a wide distribution of rTSA prostheses designs are commercially available [80]. 

## 4. rTSA Implant Positioning Considerations on Biomechanics

Irrespective of the specific rTSA design classification, rTSA biomechanics can also be influenced by modifying implant positioning. Regarding the humerus, implanting the rTSA humeral prosthesis with less humeral retroversion asymmetrically tensions the rotator cuff muscles, by increasing posterior rotator cuff tension while decreasing anterior rotator cuff tension [8]. Similarly, implanting the rTSA humeral prosthesis with more retroversion increases anterior rotator cuff tension while decreasing posterior rotator cuff tension [8]. Inadequate anterior rotator cuff tensioning may result in inadequate strength for active internal rotation; whereas, inadequate posterior rotator cuff tensioning may result in inadequate strength for active external rotation. A surgeon’s decision to implant a humeral prosthesis in more or less retroversion may be helpful for some patients as a particular patient may have a preferential need for more internal or more external rotation. Regarding the glenoid, various recommendations for glenoid baseplate positioning have been recommended to avoid scapular notching. Nyffeler et al. [81] recommended inferiorly positioning baseplate with or without an inferior tilt to reduce scapular notching. However, implanting the baseplate more inferiorly on the glenoid further elongates the deltoid, and that has implications on acromial and scapular stress fractures. Boileau et al. [76] recommended use of bone graft placed behind the baseplate with nonworn glenoids (e.g., BIO-RSA) to lateralize the humerus to better tension the rotator cuff and reduce scapular notching. However, this BIO-RSA technique lateralizes the CoR, which decreases the deltoid abductor moment arm and reduces deltoid efficiency. The BIO-RSA technique also increases the torque at the bone–implant interface, which increases the risk of glenoid loosening and introduces new complications of graft resorption and graft fracture [46,76,82,83].

## 5. rTSA Patient-Specific Considerations on Biomechanics

rTSA biomechanics can be influenced by patient-specific anatomic and morphological parameters. Every patient has different bone morphology and different muscle size/volume; as such, each patient has different muscle moment arm lengths. Jacobson et al. [16] conducted an anatomic study of 75 shoulders to investigate the variability in joint relationships between the male and female shoulders and demonstrated that middle deltoid moment arms change with gender, with male shoulders having significantly larger deltoid moment arms than those of female shoulders. The shape of the underlying bones determines how the shoulder muscles wrap around the proximal humeral anatomy, guiding deltoid wrapping and the deltoid’s line of action at various joint positions. Deltoid wrapping changes with greater tuberosity size, acromion size, acromial overhang, humeral head size, and humeral head offset. As the middle and posterior heads of the deltoid originate on the acromion and scapular spine, acromion size can also influence deltoid moment arm lengths, the amount of deltoid wrapping around the greater tuberosity, and the deltoid’s line of action at various joint positions. Concerning this point, Jacobson et al. [16] reported that acromion size and width were significantly different between male and female scapulae. For all these reasons, each patient has differing muscular capacity to generate the torque required for motion and support external loads, and the biomechanical advantages associated with rTSA may be more beneficial for some patients than for others. Furthermore, differences in bone morphology can result in differences in impingement and range of motion; for example, Middernacht et al. [84], Paisley et al. [85], and Roche et al. [86] demonstrated that shorter scapular neck lengths and larger scapular neck angles reduce rTSA range of motion and increase the risk of scapular notching [84,85,86]. 

rTSA biomechanics can also be influenced by glenoid bone loss. Glenoid wear and associated bone loss medialize the joint line, which shortens the rotator cuff muscles and reduces deltoid wrapping around the greater tuberosity [70]. Norris et al. [87] demonstrated that with a sufficient amount of medial glenoid wear, the deltoid can actually generate a laterally directed distraction force that can result in joint instability. Augmented glenoid baseplates were designed to conserve glenoid bone, increase prosthesis surface contact area with cortical bone, and better restore the native joint line when performing rTSA in eroded scapular morphologies [41,44,45]. Positive clinical results have been reported with augmented baseplates in a variety of types of eroded glenoids [83,88,89,90,91]. Alternatively, bone grafting the glenoid can be used to lateralize the joint line. However, Jones et al. [83] compared the rTSA clinical outcomes between bone grafting and augmented baseplates and reported that bone graft patients had significantly higher complication rates. 

Finally, rTSA biomechanics can be influenced by humeral bone loss. Edwards et al. [72] reported several cases of instability in rTSA patients with proximal humeral bone loss. With proximal humeral bone loss, the lesser tuberosity can be compromised, which prevents concomitant repair of the subscapularis, increasing the risk of instability for some rTSA designs, particularly an MG/MH prosthesis. Furthermore, with proximal humeral bone loss, the greater tuberosity can be compromised, which both shortens the deltoid abductor moment arm lengths to decrease deltoid efficiency and also reduces deltoid wrapping to increase risk of instability. Sabesan et al. [92] recently reported that loss of the greater tuberosity with rTSA negatively impacts shoulder biomechanics by shortening deltoid moment arms and requiring greater muscle forces during external rotation as compared to rTSA shoulders with a greater tuberosity. For patients with humeral bone loss, humeral lateralization can be recreated through the use of a novel rTSA humeral tray that includes a tuberosity augment to reconstruct the greater tuberosity of the proximal humerus to maintain a lateral deltoid over the range of motion and increase deltoid wrapping to improve stability (Figure 10). 

## 6. Conclusions

The rTSA prosthesis is geometrically different than the native glenohumeral joint, and these geometric differences impart several biomechanical advantages relative to the native shoulder and also a few disadvantages. The primary biomechanical advantage of the rTSA prosthesis is its larger deltoid abductor moment arms, which improves deltoid efficiency, and this, as conceived by Grammont, is achieved by medializing the CoR relative to the native shoulder. Building on this heritage, modern rTSA prostheses have been demonstrated to be an effective treatment solution to restore stability and function for patients with a wide variety of end-stage degenerative conditions of the shoulder. However, some differences in outcomes and complications have been observed between rTSA prostheses, and these differences can be partially explained by the biomechanical impact associated with the different glenoid and humeral prosthesis design parameters, and also by the impact of surgical technique, implant positioning, patient-specific bone morphology, and usage in humeral and glenoid bone loss situations on rTSA biomechanics. Which prosthesis to select for a given patient or surgical situation is not always known and may not be obvious, and future work may utilize machine learning analytic techniques, as recently performed by Kumar et al., to elucidate previously unknown correlations related to prosthesis design, surgical technique, implant placement, and implant size/type selection to better predict clinical outcomes and optimize clinical decision-making [93,94,95,96,97]. Ultimately, a better understanding of the biomechanical principles associated with rTSA prosthesis design and also surgical technique can help orthopedic surgeons improve clinical decision-making; furthermore, this knowledge should guide future innovations and further improve rTSA clinical outcomes while also reducing complication rates. 

## Figures and Tables

**Figure 1 jfmk-07-00013-f001:**
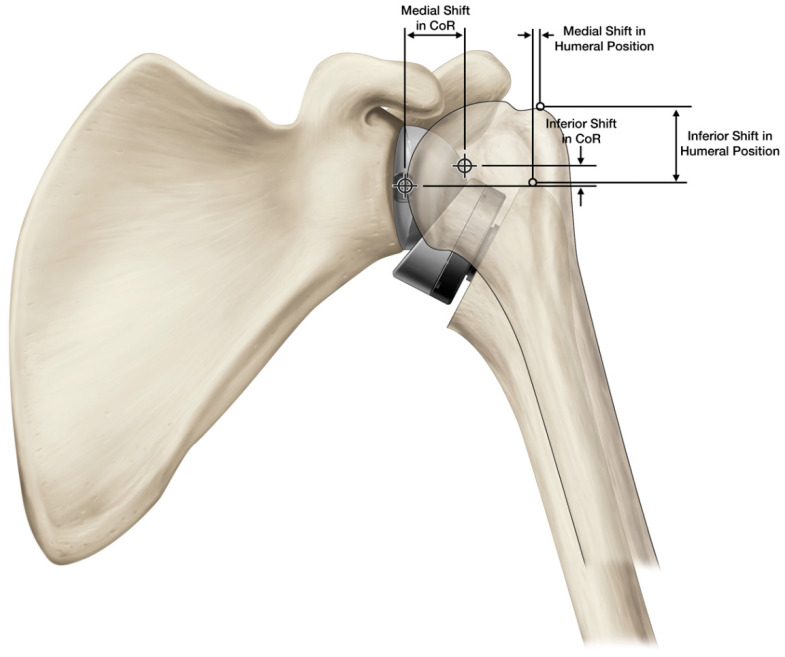
Inferior and medial translation of the CoR and humeral position with rTSA, relative to the native shoulder.

**Figure 2 jfmk-07-00013-f002:**
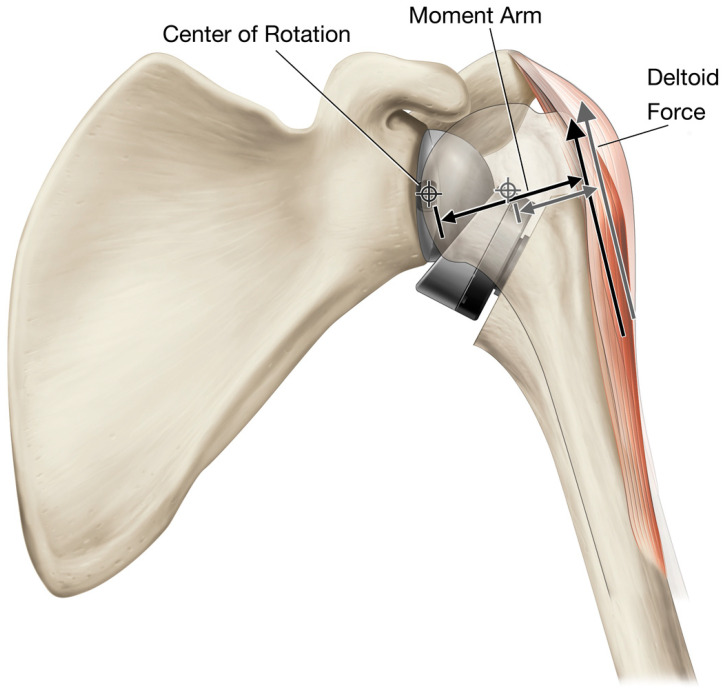
Increase in deltoid moment arm length with rTSA, relative to the native shoulder.

**Figure 3 jfmk-07-00013-f003:**
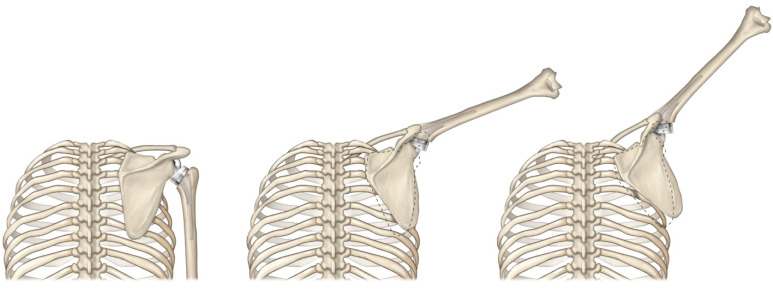
Modified scapulohumeral rhythm with rTSA, where additional scapular rotation occurs with rTSA relative to the scapulohumeral rhythm of the native shoulder (dotted line) during arm elevation.

**Figure 4 jfmk-07-00013-f004:**
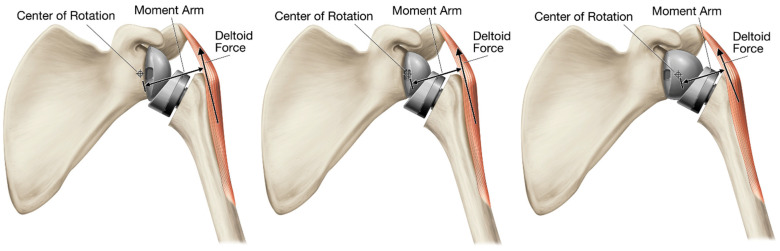
Increase in deltoid abductor moment arm length by medializing the CoR: inset CoR glenosphere (**left**), standard offset glenosphere (**middle**), and expanded glenosphere (**right**).

**Figure 5 jfmk-07-00013-f005:**
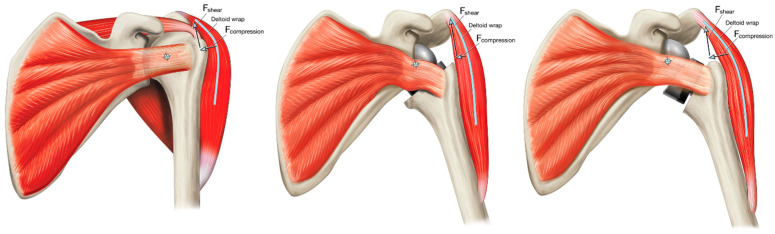
Wrapping of the middle deltoid around the lateral proximal humerus generates a stabilizing compressive force; where a greater amount of deltoid wrapping results in a larger compression vector that imparts greater joint stability: native shoulder (**left**), medial glenoid/medial humerus design (**middle**), and medial glenoid/lateral humerus design (**right**).

**Figure 6 jfmk-07-00013-f006:**
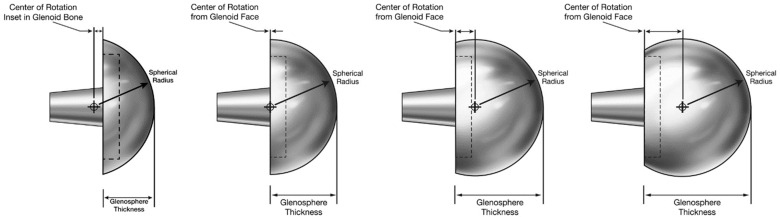
rTSA glenoid prosthesis design classification, representative images of four glenosphere designs having equivalent articular curvatures, demonstrating that the relationship between glenoid thickness and articular radius is directly related to the lateralization of the CoR relative to the glenoid fossa.

**Figure 7 jfmk-07-00013-f007:**
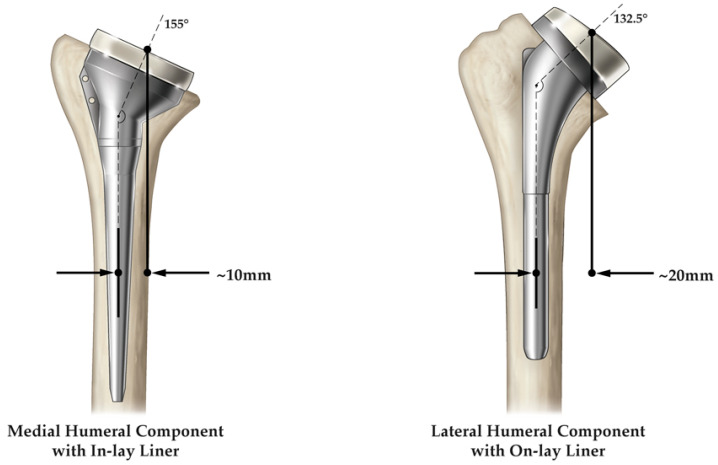
rTSA humeral prosthesis design classification, examples of a medial humeral component with an inlay humeral liner (**left**) and a lateral humeral component with an onlay humeral liner (**right**).

**Figure 8 jfmk-07-00013-f008:**
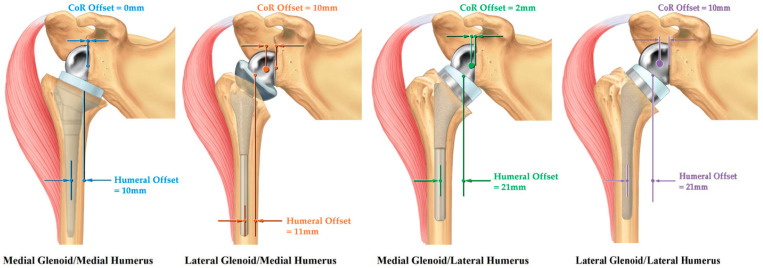
rTSA prosthesis design classification.

**Figure 9 jfmk-07-00013-f009:**
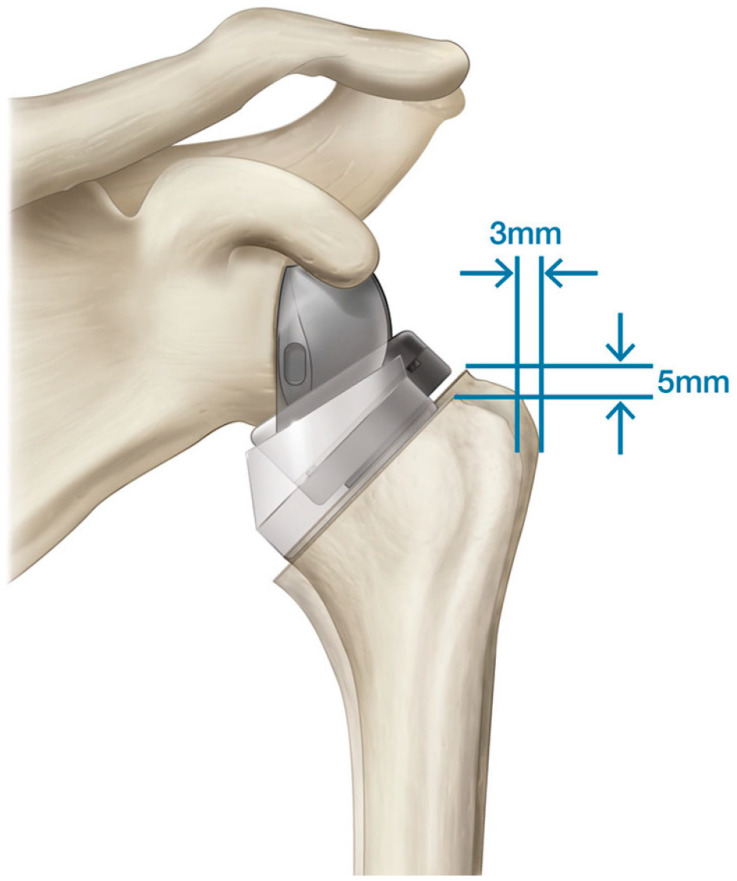
Comparison of lateral and distal humeral positioning associated with two different MGLH reverse shoulder prosthesis designs.

**Figure 10 jfmk-07-00013-f010:**
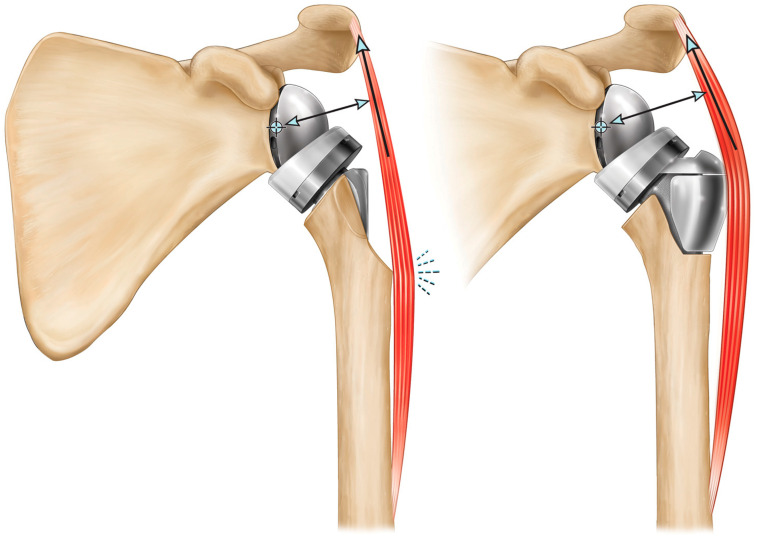
Use of an augmented humeral tray with rTSA in the case of proximal humeral bone loss to improve rTSA biomechanics by restoring the greater tuberosity shape and increasing deltoid wrapping to improve stability while increasing the deltoid moment arm to improve deltoid efficiency.

## Data Availability

Not applicable.

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
