# Peer review of "Reverse Shoulder Arthroplasty Biomechanics"

_jfmk, 2022, doi:10.3390/jfmk7010013_

Round 1

Reviewer 1 Report

Dear author!

I liked reading your paper about rTSA. I think the topic is worked up in a great way, and I really like the pictures - for better understanding.

A minor spell check is needed. 

After this I would recommend publication.

With kind regards.

Author Response

Thank you for your review; I appreciate the kind words and you taking the time to review this work. As you recommended I conducted a spell-check and found no misspellings, I did make a few minor changes based upon the feedback from the other reviewers. Thank you again 

Reviewer 2 Report

good review indeed, not sure adding anything to current knowledge but for sure a nice up to date of the present 

Author Response

Thank you for your review and kind words; I appreciate you taking the time to review this work. I made a few minor edits based upon the feedback of the other reviewers. Thanks again!

Reviewer 3 Report

Interesting article on the biomechanics of shoulder prostheses.

I have a few comments:

Abstract:

Please explain the abbreviations: CoR and rTSA.

Conclusion

I am asking for some clear conclusions for orthopedic surgeons about the benefits of rTSA and when it is best to use it.

Author Response

Thank you for your review. As you recommended I spelled out the meaning of the abbreviations for rTSA and CoR in the abstract and also the first usage in the text. Also, as you recommended, I rewrote the final sentence of the conclusion section to provide more relevance to the orthopedic surgeon, as follows: "Ultimately, a better understanding of the biomechanical principles associated with rTSA prosthesis design and also surgical technique can help orthopedic surgeons improve clinical decision making; furthermore, this knowledge should guide future innovations and further improve rTSA clinical outcomes while also reduce complication rates. " I appreciate you taking the time to review this work.  

Round 2

Reviewer 2 Report

Thank you no more corrections needed